



# Analytical solution for viscous incompressible Stokes flow in a spherical shell

Cedric Thieulot[1]

[1]Department of Earth Sciences, Utrecht University, Utrecht, Netherlands

*Correspondence to:* C. Thieulot (c.thieulot@uu.nl)

**Abstract.** I present a new family of analytical flow solutions to the incompressible Stokes equation in a spherical shell. The velocity is tangential to both inner and outer boundaries, the viscosity is radial and of power-law type, and the solution has been designed so that the expressions for velocity, pressure, and body force are simple polynomials and therefore simple to implement in (geodynamics) codes. Various flow average values, e.g. the root mean square velocity, are analytically computed. This forms the basis for a numerical benchmark for convection codes and I have implemented it in two finite element codes ASPECT and ELEFANT. I report on error convergence rates for velocity and pressure.



# 1  Introduction

With the advent of ever more refined modelling techniques and powerful computers, numerical modelling of mantle convection has grown in complexity since the mantle is a very complex region characterised by large variations in temperature, viscosity, composition, phase changes , melting (e.g. van Heck et al. (2016); Dannberg and Heister (2016)) and anisotropic structures as revealed by seismic data (Tackley, 2012).

Also it has been recognised that the mantle exerts a primary control on the evolution of tectonic plates and that both should be simulated together if one is to build a fully dynamic Earth model (e.g. van Hinsbergen et al. (2011), Bull et al. (2014), Bower et al. (2015)).

Many codes have seen the light of day in the last thirty years (Zhong et al., 2000; Choblet et al., 2007; Tackley, 2008; Davies et al., 2013; Kronbichler et al., 2012; Burstedde et al., 2013), and a handful of spherical shell numerical benchmarks have been carried out (Stemmer et al., 2006; Zhong et al., 2008; Arrial et al., 2014).

Semi-analytical Stokes flow solutions derived via propagator matrix methods have also been proposed in the past (Busse, 1975; Busse and Riahi, 1982; Hager and O'Connell, 1981; Richards and Hager, 1984) while Tosi and Martinec (2007) derived a semi-analytical solution in the case of two eccentrically nested spheres. Fully analytical solutions have also been recently proposed, attempting to represent a mid-ocean ridge (Burstedde et al., 2013) or being more abstract in nature (Blinova et al., 2016).

While inter-code comparisons are useful, the above mentioned benchmark studies rely on the comparisons between a handful of scalar values (e.g. root mean square velocity, Nusselt number) which account for the global character of the solution but they do not lend themselves to error convergence measurements since the analytical solution is not known.

Also, the semi-analytical solutions present a major drawback: the solution is given as a function of spherical harmonic expansions which are based on infinite sums and which can be cumbersome to manipulate and/or implement.

Finally, the existing 'pure' analytical solutions do not satisfy the condition $v \cdot n = 0$ on the inner and outer boundaries, i.e. there is flow through the boundaries. Not only does the presence of a material flow through the boundares makes the flow not Earth-like, but it also precludes its use for particle-in-cell advection benchmarking (particles would leave and enter the domain).

I here propose a new analytical solution for viscous incompressible Stokes flow in a spherical shell. It has been designed with three constraints in mind: 1) the boundary conditions, buoyancy forces and viscosity fields should be simple to implement and exact; 2) the solution should also be smooth and simple enough to be usable; 3) it should satisfy tangential slip boundary conditions on both surfaces.

I present in section 2 the simplified Stokes equation in spherical coordinates under these flow assumptions, and outline the procedure to arrive at the analytical solution for pressure and velocity in section 3, for both constant and depth-dependent viscosity profiles. I compute in section 4 the exact analytical values for the root mean square velocity and various other averages and in section 5 the two numerical codes are introduced and results obtained with these are shown. Finally, these results are discussed in section 6.





## 2    The Stokes equations in spherical coordinates

The domain is a spherical shell parametrised by its inner radius $R_1$ and outer radius $R_2$. For an incompressible fluid, the Stokes flow equations are given by

$$\boldsymbol{\nabla} \cdot \boldsymbol{v} \;\; = \;\; 0 \tag{1}$$

20
$$\boldsymbol{\nabla} \cdot \boldsymbol{\sigma} + \rho \boldsymbol{g} \;\; = \;\; \boldsymbol{0} \tag{2}$$

where $\boldsymbol{v}$ is the velocity vector, $\rho$ the mass density, $\boldsymbol{g}$ the gravitational acceleration vector and where $\boldsymbol{\sigma}$ is the full stress tensor which can be split

$$\boldsymbol{\sigma} = -p\mathbf{1} + \boldsymbol{s} \tag{3}$$

where $p$ is the pressure, $\mathbf{1}$ the unit tensor and $\boldsymbol{s}$ the deviatoric stress tensor. Eq. (2) then becomes

25
$$-\boldsymbol{\nabla} p + \boldsymbol{\nabla} \cdot \boldsymbol{s} + \rho \boldsymbol{g} \;\; = \;\; 0. \tag{4}$$

In spherical coordinates, equations (1) and (4) become:

$$\frac{1}{r^2}\frac{\partial}{\partial r}(r^2 v_r) + \frac{1}{r\sin\theta}\frac{\partial}{\partial \theta}(v_\theta \sin\theta) + \frac{1}{r\sin\theta}\frac{\partial v_\phi}{\partial \phi} \;\; = \;\; 0 \tag{5}$$

$$-\frac{\partial p}{\partial r} + \frac{\partial s_{rr}}{\partial r} + \frac{1}{r}\frac{\partial s_{\theta r}}{\partial \theta} + \frac{1}{r\sin\theta}\frac{\partial s_{\phi r}}{\partial \phi} + \frac{2s_{rr} - s_{\theta\theta} - s_{\phi\phi}}{r} + \frac{s_{\theta r}\cot\theta}{r} + \rho g_r \;\; = \;\; 0 \tag{6}$$

$$-\frac{1}{r}\frac{\partial p}{\partial \theta} + \frac{\partial s_{r\theta}}{\partial r} + \frac{1}{r}\frac{\partial s_{\theta\theta}}{\partial \theta} + \frac{1}{r\sin\theta}\frac{\partial s_{\phi\theta}}{\partial \phi} + \frac{3s_{\theta r} + (s_{\theta\theta} - s_{\phi\phi})\cot\theta}{r} + \rho g_\theta \;\; = \;\; 0 \tag{7}$$

$$-\frac{1}{r\sin\theta}\frac{\partial p}{\partial \phi} + \frac{\partial s_{r\phi}}{\partial r} + \frac{1}{r}\frac{\partial s_{\theta\phi}}{\partial \theta} + \frac{1}{r\sin\theta}\frac{\partial s_{\phi\phi}}{\partial \phi} + \frac{3s_{r\phi} + 2s_{\phi\theta}\cot\theta}{r} + \rho g_\phi \;\; = \;\; 0 \tag{8}$$

In this work, the following spherical coordinates conventions are used: $r$ is the radial distance, $\theta \in [0, \pi]$ is the polar angle and $\phi \in [0, 2\pi]$ is the azimuthal angle. In the case of an incompressible fluid, the deviatoric stress is simply

$$\boldsymbol{s} = 2\mu\dot{\boldsymbol{\epsilon}} \tag{9}$$

where $\mu$ is the dynamic viscosity which can depend on space coordinates and $\dot{\boldsymbol{\epsilon}}$ is the (deviatoric) strain rate tensor:

$$\dot{\boldsymbol{\epsilon}} = \frac{1}{2}\left(\boldsymbol{\nabla}\boldsymbol{v} + \boldsymbol{\nabla}\boldsymbol{v}^T\right). \tag{10}$$



In spherical coordinates, the components of the deviatoric stress tensor are given by:

$$s_{rr} = 2\mu \left( \frac{\partial v_r}{\partial r} \right) \tag{11}$$

$$s_{\theta\theta} = 2\mu \left( \frac{1}{r} \frac{\partial v_\theta}{\partial \theta} + \frac{v_r}{r} \right) \tag{12}$$

$$s_{\phi\phi} = 2\mu \left( \frac{1}{r\sin\theta} \frac{\partial v_\phi}{\partial \phi} + \frac{v_r}{r} + \frac{v_\theta}{r} \cot\theta \right) \tag{13}$$

$$s_{r\theta} = s_{\theta r} = \mu \left( \frac{\partial v_\theta}{\partial r} - \frac{v_\theta}{r} + \frac{1}{r} \frac{\partial v_r}{\partial \theta} \right) \tag{14}$$

$$s_{r\phi} = s_{\phi r} = \mu \left( \frac{1}{r\sin\theta} \frac{\partial v_r}{\partial \phi} + \frac{\partial v_\phi}{\partial r} - \frac{v_\phi}{r} \right) \tag{15}$$

$$s_{\theta\phi} = s_{\phi\theta} = \mu \left( \frac{1}{r} \frac{\partial v_\phi}{\partial \theta} - \frac{\cot\theta\, v_\phi}{r} + \frac{1}{r\sin\theta} \frac{\partial v_\theta}{\partial \phi} \right). \tag{16}$$

Equations 5-8, supplemented by Eqs. 11-16 form a closed set of PDE's which can be solved given an appropriate set of boundary conditions.

## 2.1 Assumptions on the flow

In order to derive an analytical solution for the flow velocity and pressure in the domain, the following assumptions are made:

- All quantities $v_r, v_\theta, v_\phi, p, \rho$ and $\mu$ are independent of the azimuthal angle $\phi$. As a consequence, all the terms containing partial derivatives with respect to $\phi$ can be discarded. This is one of the most stringent limitations in this work since it implies rotational symmetry with respect to the vertical axis.

- The polar and azimuthal components of the velocity are equal and of the form

$$v_\theta(r,\theta) = v_\phi(r,\theta) = f(r)\sin\theta. \tag{17}$$

- The radial component of the velocity is nul on the inside $r = R_1$ and outside $r = R_2$ of the domain, thereby insuring a tangential flow on the boundaries, i.e.

$$v_r(R_1,\theta) = v_r(R_2,\theta) = 0. \tag{18}$$

- The viscosity is a function of the radial distance only and takes the form:

$$\mu(r) = \mu_0 r^{m+1}. \tag{19}$$

where $m$ is an integer (positive or negative). Note that $m = -1$ yields a constant viscosity.

- The gravity vector is set to $\boldsymbol{g} = -\boldsymbol{e}_r$ and is therefore of unit norm, i.e. $|\boldsymbol{g}| = 1$.



### 2.1.1 The set of simplified partial differential equations

Under the above assumptions, it is easy to show that the Stokes equation in spherical coordinates are given by:

$$\frac{1}{r^2}\frac{\partial}{\partial r}(r^2 v_r) + \frac{1}{r\sin\theta}\frac{\partial}{\partial\theta}(v_\theta\sin\theta) = 0 \tag{20}$$

$$-\frac{\partial p}{\partial r} + \rho(r,\theta) + \mu(r)\left(\Delta v_r - \frac{2v_r}{r^2} - \frac{2}{r^2}\frac{\partial v_\theta}{\partial\theta} - \frac{2v_\theta\cot\theta}{r^2}\right) + 2\mu'(r)\frac{\partial v_r}{\partial r} = 0 \tag{21}$$

$$-\frac{1}{r}\frac{\partial p}{\partial\theta} + \mu(r)\left(\Delta v_\theta + \frac{2}{r^2}\frac{\partial v_r}{\partial\theta} - \frac{v_\theta}{r^2\sin^2\theta}\right) + \mu'(r)\left(\frac{\partial v_\theta}{\partial r} - \frac{v_\theta}{r} + \frac{1}{r}\frac{\partial v_r}{\partial\theta}\right) = 0 \tag{22}$$

$$\mu(r)\left(\Delta v_\phi - \frac{v_\phi}{r^2\sin^2\theta}\right) + \mu'(r)\left(\frac{\partial v_\phi}{\partial r} - \frac{v_\phi}{r}\right) = 0 \tag{23}$$

where $\Delta$ is the Laplacian operator:

$$\Delta = \frac{1}{r^2}\frac{\partial}{\partial r}\left(r^2\frac{\partial}{\partial r}\right) + \frac{1}{r^2\sin\theta}\frac{\partial}{\partial\theta}\left(\sin\theta\frac{\partial}{\partial\theta}\right) + \frac{1}{r^2\sin^2\theta}\frac{\partial^2}{\partial\phi^2}. \tag{24}$$

## 3 Derivation of the analytical solution of the Stokes equations

My goal is to determine the $f(r)$ function in the context of all the assumptions made about the flow nature. The following 4 steps will then be taken:

1. use the continuity equation (20) to arrive at $v_r(r,\theta) = g(r)\cos\theta$;

2. use the $\theta$ component of Stokes equations (22) to arrive at $p(r,\theta) = h(r)\cos\theta$;

3. use the $\phi$ component of Stokes equations (23) to arrive at $f(r)$ and $g(r)$, using boundary conditions on $v_r$;

4. use the $r$ component of Stokes equations (21) to arrive at the density $\rho(r,\theta)$.

### 3.1 Using the continuity equation to arrive at $v_r$

Inserting Eq. (17) into Eq. (20) yields

$$v_r(r,\theta) = g(r)\cos\theta \tag{25}$$

$$g(r) = -\frac{2}{r^2}\int r f(r)dr \tag{26}$$

where the integration constant has been set to zero for simplicity.

I then proceed to compute the Laplacian of each velocity component using Eq. (24) and once again all partial derivatives with respect to $\phi$ are neglected:



$$\Delta v_r \;=\; \left[ g'' + \frac{2g'}{r} - 2\frac{g}{r^2} \right] \cos\theta \tag{27}$$

$$\Delta v_\theta \;=\; \left( f'' + \frac{2f'}{r} \right) \sin\theta + \frac{f}{r^2 \sin\theta} \left( \cos^2\theta - \sin^2\theta \right) \tag{28}$$

$$\Delta v_\phi \;=\; \left( f'' + \frac{2f'}{r} \right) \sin\theta + \frac{f}{r^2 \sin\theta} \left( \cos^2\theta - \sin^2\theta \right). \tag{29}$$

### 3.2 Using the $\theta$ component of Stokes equations to arrive at the pressure $p$

Inserting Eq. (17) and Eq. (25) into Eq. (22) yields

$$\frac{\partial p}{\partial \theta} = -h(r)\sin\theta \tag{30}$$

with

$$h(r) = -\mu(2f' + rf'') + \frac{2\mu + r\mu'}{r}(f + g) - r\mu'f' \tag{31}$$

so that

$$p(r,\theta) = h(r)\cos\theta \tag{32}$$

where the integration constant has once again been omitted for simplicity.

### 3.3 Using the $\phi$ component of Stokes equations to arrive at $f(r)$ and $g(r)$

Inserting Eq.(17) and Eq.(25) into Eq. (23) yields

$$\mu \left( r^2 f'' + 2rf' - 2f \right) + r\mu' \left( rf' - f \right) \;=\; 0 \tag{33}$$

We now make use of Eq.(19) so that Eq.(33) becomes

$$r^{m+1} \left[ r^2 f'' + (3+m)rf' - (3+m)f \right] = 0 \tag{34}$$

This equation has to hold for all $r$ values so $f(r)$ is the solution of the second order ODE:

$$r^2 f'' + (3+m)rf' - (3+m)f = 0 \tag{35}$$

I postulate that the solution is of the form $f(r) = r^a$ which yields

$$[a + (m+3)](a-1)f(r) = 0 \tag{36}$$

The only two acceptable values for $a$ are the roots of this second order polynomial, i.e. $a = 1$ or $a = -(m+3)$. The general solution of the ODE is then:

$$f(r) = \alpha r^{-(m+3)} + \beta r \tag{37}$$





where $\alpha$ and $\beta$ are two constants yet to be determined. Having obtained $f(r)$, one can now compute $g(r)$. However, as it will become obvious, one must make a distinction between $m = -1$ and $m \neq -1$.

Note that the case $m = -3$ should be considered separately since Eq.(35) then becomes $f''(r) = 0$ and yields a solution $f(r) = \beta r$.

### 3.3.1  Case $m = -1$

In this case the function $f(r)$ is

$$f(r) = \frac{\alpha}{r^2} + \beta r \tag{38}$$

and from Eq. (26) one obtains

$$g(r) = -\frac{2}{r^2}\left(\alpha \ln r + \frac{\beta}{3}r^3 + \gamma\right) \tag{39}$$

where $\gamma \neq 0$ is a constant. From Eq. (18) and Eq. (25) it follows that $g(R_1) = g(R_2) = 0$ which yields

$$\alpha = -\gamma\frac{R_2^3 - R_1^3}{R_2^3 \ln R_1 - R_1^3 \ln R_2} \tag{40}$$

$$\beta = -3\gamma\frac{\ln R_2 - \ln R_1}{R_1^3 \ln R_2 - R_2^3 \ln R_1}. \tag{41}$$

### 3.3.2  Case $m \neq -1$

In this case the function $g(r)$ takes the form

$$g(r) = -\frac{2}{r^2}\left(-\frac{\alpha}{m+1}r^{-(m+1)} + \frac{\beta}{3}r^3 + \gamma\right) \tag{42}$$

and the boundary conditions impose that

$$\alpha = \gamma(m+1)\frac{R_1^{-3} - R_2^{-3}}{R_1^{-(m+4)} - R_2^{-(m+4)}} \tag{43}$$

$$\beta = -3\gamma\frac{R_1^{m+1} - R_2^{m+1}}{R_1^{m+4} - R_2^{m+4}} \tag{44}$$

Note that this imposes that $m \neq -4$.

### 3.4  Using the $r$ component of Stokes equations to arrive at density $\rho$

Eq. (21) contains the term $\partial p/\partial r = \partial h/\partial r \cos\theta$ which needs to be addressed beforehand:

$$\frac{\partial h}{\partial r} = -\mu'(2f' + rf'') - \mu(3f'' + rf''') + \frac{r^2\mu'' + 2r\mu' - 2\mu}{r^2}(f+g) + \frac{2\mu + r\mu'}{r}(f' + g') - \mu'f' - r\mu''f' - r\mu'f''. \tag{45}$$





### 3.4.1 Case $m = -1$

In this case $\mu' = 0$ and $\mu'' = 0$ so that $\partial h / \partial r \cos \theta$ simplifies to

$$\frac{\partial h}{\partial r} = -(3f'' + rf''') + \frac{2}{r}(f' + g') - \frac{2}{r^2}(f + g) \tag{46}$$

and Eq. (21) becomes

$$-\frac{\partial h}{\partial r} + \frac{\rho(r,\theta)}{\cos \theta} + \frac{\Delta v_r}{\cos \theta} - \frac{1}{r^2}(2g + 4f) = 0. \tag{47}$$

I postulate that

$$\rho(r,\theta) = \mathcal{F}(r) \cos \theta \tag{48}$$

and using Eq. (27), the radial function $\mathcal{F}$ is given by

$$\mathcal{F}(r) = -rf''' - 3f'' + 2\frac{f'}{r} - g'' + \frac{2}{r^2}(f + g). \tag{49}$$

Inserting the radial functions $f(r)$ and $g(r)$ given into Eqs. (38) and (39) into Eq. (49) yields

$$\rho(r,\theta) = \left( \frac{\alpha}{r^4}(8\ln r - 6) + \frac{8\beta}{3r} + 8\frac{\gamma}{r^4} \right) \cos \theta. \tag{50}$$

### 3.4.2 Case $m \neq -1$

Using Eq. (19) leads to write

$$\frac{1}{r^m} \frac{\partial h}{\partial r} = -r^2 f''' - [2m + 5]rf'' - m(m + 3)f' + m(m + 3)(f + g)/r + (m + 3)g'. \tag{51}$$

Then Eq. (21) becomes

$$-\frac{\partial h}{\partial r} \cos \theta + \rho(r,\theta) + \mu(r)\left[\Delta v_r - \frac{\cos \theta}{r^2}(2g + 4f)\right] + \mu'(r)g'(r)\cos \theta = 0 \tag{52}$$

I here postulate that

$$\rho(r,\theta) = r^m \mathcal{F}(r) \cos \theta \tag{53}$$

and arrive at

$$\mathcal{F}(r) = \frac{1}{r^m} \frac{\partial h}{\partial r} - r\left[g'' + \frac{2(m + 2)}{r}g' - \frac{4}{r^2}(g + f)\right] \tag{54}$$

$$= -r^2 f''' - [2m + 5]rf'' - m(m + 3)f' + [m(m + 3) + 4]\frac{f + g}{r} - (m + 1)g' - rg'' \tag{55}$$

$$= -r^2 f''' - [2m + 5]rf'' - [m(m + 3) - 2]f' + m(m + 5)\frac{f + g}{r} \tag{56}$$

where I have used

$$g'(r) = -\frac{2}{r}(f + g) \tag{57}$$

$$rg''(r) = -2f' + \frac{6}{r}(f + g). \tag{58}$$





Inserting $f(r)$ and $g(r)$ expressions into the above equation yields $\mathcal{F}(r)$ so that in the end

$$\rho(r,\theta) = \left[ 2\alpha r^{-(m+4)} \frac{m+3}{m+1}(m-1) - \frac{2\beta}{3}(m-1)(m+3) - m(m+5)\frac{2\gamma}{r^3} \right] \cos\theta. \tag{59}$$

### 3.5 The pressure field

The pressure is defined in Eq. (32) and $h(r)$ can now be computed.

#### 3.5.1 Case $m = -1$

$$h(r) = -\mu_0 \frac{4}{r^3} \left( \alpha \ln r + \frac{\beta}{3}r^3 + \gamma \right) = \frac{2}{r}\mu_0 g(r). \tag{60}$$

#### 3.5.2 Case $m \neq -1$

$$h(r) = -\mu(r)\frac{2(m+3)}{r^3}\left( -\frac{\alpha}{m+1}r^{-(m+1)} + \frac{\beta}{3}r^3 + \gamma \right) = \frac{m+3}{r}\mu(r)g(r). \tag{61}$$

## 4 Additional measurements

### 4.1 Root mean square velocity

Many benchmark studies (e.g. Blankenbach et al. (1989); Tosi et al. (2015)) report on the root mean square velocity quantity,

defined as follows:

$$v_{rms} = \sqrt{\frac{1}{V}\int_V |v|^2 dV} \tag{62}$$

This is a convenient quantity as it captures (in an average sense) the nature of the velocity field in a single scalar value which allows for an easy comparison, either with a known analytical value or across multiple codes.

Since the velocity is known in all of the domain, it is a simple although tedious exercise to compute the rms velocity. I find:

$$v_{rms} = \sqrt{\frac{4\pi}{3V}[B+4A]} \tag{63}$$

with

$$A = \int_{R_1}^{R_2} f^2 r^2 dr \tag{64}$$

$$B = \int_{R_1}^{R_2} g^2 r^2 dr. \tag{65}$$

The values of $A$ and $B$ depend on the $f$ and $g$ function, so that we must once again make the distinction between $m = -1$ and $m \neq -1$.





### 4.1.1 Case $m = -1$

$$A = \left[ -\frac{\alpha^2}{r} + \alpha\beta r^2 + \frac{\beta^2}{5}r^5 \right]_{R_1}^{R_2} \tag{66}$$

$$B = 4(B_1 + B_2 + B_3 + B_4 + B_5 + B_6) \tag{67}$$

$$B_1 = \alpha^2 \left[ -(X^2 + 2X + 2)e^{-X} \right]_{\ln R_1}^{\ln R_2} \tag{68}$$

$$B_2 = \frac{2\alpha\beta}{3} \left[ \frac{1}{2}r^2 \ln r - \frac{1}{4}r^2 \right]_{R_1}^{R_2} \tag{69}$$

$$B_3 = 2\alpha\gamma \left[ -Xe^{-X} - e^{-X} \right]_{\ln R_1}^{\ln R_2} \tag{70}$$

$$B_4 = \frac{\beta^2}{45} \left[ r^5 \right]_{R_1}^{R_2} \tag{71}$$

$$B_5 = \frac{\beta\gamma}{3} \left[ r^2 \right]_{R_1}^{R_2} \tag{72}$$

$$B_6 = \gamma^2 \left[ -\frac{1}{r} \right]_{R_1}^{R_2} \tag{73}$$

### 4.1.2 Case $m \neq -1$

$$A = \left[ -\frac{\alpha^2}{2m+3}r^{-(2m+3)} - \frac{2\alpha\beta}{m-1}r^{-(m-1)} + \frac{\beta^2}{5}r^5 \right]_{R_1}^{R_2} \tag{74}$$

$$B = 4(B_1 + B_2 + B_3 - B_4 - B_5 + B_6) \tag{75}$$

$$B_1 = -\frac{\alpha^2}{(m+1)^2(2m+3)} \left[ r^{-(2m+3)} \right]_{R_1}^{R_2} \tag{76}$$

$$B_2 = \frac{\beta^2}{45} \left[ r^5 \right]_{R_1}^{R_2} \tag{77}$$

$$B_3 = -\gamma^2 \left[ \frac{1}{r} \right]_{R_1}^{R_2} \tag{78}$$

$$B_4 = -\frac{2\alpha\beta}{3(m+1)(m-1)} \left[ r^{-(m-1)} \right]_{R_1}^{R_2} \tag{79}$$

$$B_5 = -\frac{2\alpha\gamma}{(m+1)(m+2)} \left[ r^{-(m+2)} \right]_{R_1}^{R_2} \tag{80}$$

$$B_6 = \frac{\beta\gamma}{3} \left[ r^2 \right]_{R_1}^{R_2} \tag{81}$$

### 4.2 Radial averages

The radial average of a quantity $\chi(r, \theta, \phi)$ is defined as follows

$$< \chi >_R = \frac{1}{4\pi} \int \int \chi(r, \theta, \phi) \sin\theta d\theta d\phi \tag{82}$$

and due to symmetry it is trivial to show that $< p >_R = 0$ and $< v_r >_R = 0$. Likewise, one easily arrives at

$$< v_\theta >_R = < v_\phi >_R = \frac{1}{4}f(r). \tag{83}$$



### 4.3 Volume averages

The volume average of a quantity $\chi(r,\theta,\phi)$ is defined as follows

$$<\chi>_V = \frac{1}{V}\int_V \chi(r,\theta,\phi)dV. \tag{84}$$

Here again, $<p>_V = 0$ and $<v_r>_V = 0$ while

$$<v_\theta> = <v_\phi> = \frac{\pi^2}{V}\int_{R_1}^{R_2} f(r)r^2 dr = \begin{cases} \frac{\pi^2}{V}\left[\alpha r + \frac{\beta}{4}r^4\right]_{R_1}^{R_2} & (m=-1) \\ \frac{\pi^2}{V}\left[-\frac{\alpha}{m}r^{-m} + \frac{\beta}{4}r^4\right]_{R_1}^{R_2} & (m \neq -1) \end{cases} \tag{85}$$

One can also look at the volume averages of the cartesian coordinates components of the velocity:

$$<v_x> = \frac{1}{V}\int_V v_x(r,\theta,\phi)dV \tag{86}$$

$$= \frac{1}{V}\int_V (\sin\theta\cos\phi\, v_r + \cos\theta\cos\phi\, v_\theta - \sin\phi\, v_\phi)dV \tag{87}$$

$$= 0 \tag{88}$$

$$<v_y> = \frac{1}{V}\int_V v_y(r,\theta,\phi)dV \tag{89}$$

$$= \frac{1}{V}\int_V (\sin\theta\sin\phi\, v_r + \cos\theta\sin\phi\, v_\theta + \cos\phi\, v_\phi)dV \tag{90}$$

$$= 0 \tag{91}$$

$$<v_z> = \frac{1}{V}\int_V v_z(r,\theta,\phi)dV \tag{92}$$

$$= \frac{1}{V}\int_V (v_r\cos\theta - v_\theta\sin\theta)dV \tag{93}$$

$$= \frac{1}{V}\int_V (g(r)\cos^2\theta - f(r)\sin^2\theta)r^2\sin\theta dr d\theta d\phi \tag{94}$$

$$= \frac{4\pi}{3V}\left[\int g(r)r^2 dr - 2\int f(r)r^2 dr\right]$$

$$= 0 \tag{95}$$

Note that $<v_x>$ and $<v_y>$ are zero because of symmetry ($\int_0^{2\pi}\cos\phi d\phi = \int_0^{2\pi}\sin\phi d\phi = 0$) while we find $<v_z> = 0$ (for all values of $m$) after tedious calculations, using the definitions of $\alpha$ and $\beta$.

### 4.4 Surface averages

The average of a quantity $\chi(r,\theta,\phi)$ on a surface of radius $R$ is simply the radial average function evaluated at a given radial distance $R$. Rather importantly, we have

$$<p>_{R=R_2} = 0 \tag{96}$$





i.e., the average pressure at the outside surface is zero.

### 4.5   Moment of inertia

Because of the expression of the density field, i.e. $\rho(r,\theta,\phi) = \mathcal{F}(r)\cos\theta$, it is trivial to show that the moment of inertia of the system with respect to the $x-$, $y-$ and $z-$axis are identically zero.

### 4.6   Stress field

Since the velocity and pressure fields are known, I can also compute an analytical expression for the stress field and the the stress tensor is given by

$$\boldsymbol{\sigma} = \frac{\mu(r)}{r}\begin{pmatrix} -((m+7)g+4f)\cos\theta & (rf'-f-g)\sin\theta & (rf'-f)\sin\theta \\ (rf'-f-g)\sin\theta & -((m+1)g-2f)\cos\theta & 0 \\ (rf'-f)\sin\theta & 0 & -((m+1)g-2f)\cos\theta \end{pmatrix}$$

## 5   Implementation and results

As mentioned earlier, this flow solution was designed with a geodynamics application in mind. It has therefore been imple-
mented in the state-of-the-art open source code ASPECT[1] (Kronbichler et al., 2012; Heister et al., 2017) and in the ELEFANT[2] code (Thieulot, 2014; Tosi et al., 2015; Lavecchia et al., 2017). Both codes solve the incompresible flow Stokes equations in spherical shell domains but use Cartesian coordinates.

### 5.1   ASPECT

ASPECT is a Finite Element code intended to solve the equations that describe thermally driven convection with a focus on
doing so in the context of convection in Earth's mantle. The default element type $Q_2Q_1$ (quadratic velocity, linear pressure) has been used in this work, but since ASPECT is based on the deal.ii library (Bangerth et al., 2007, 2016), one can easily change the element type from the ascii input file ('.prm' file) and the $Q_1P_0$ (linear velocity, constant pressure) was also used (the same element is used in the CITCOM code (Zhong et al., 2008)).

I here make use of the plugin architecture of the code which allows users/developers to easily add or switch between already implemented features. The flow velocity and pressure solutions, as well as the viscosity and body force expressions are all
encapsulated in a single piece of code alongside an ascii input file in which resolution, element type, boundary conditions and other parameters are set. This benchmark is now part of the mainline since version 2.0.0-pre (see ASPECT manual).

---

[1]https://aspect.dealii.org/
[2]http://cedricthieulot.net/elefant.html





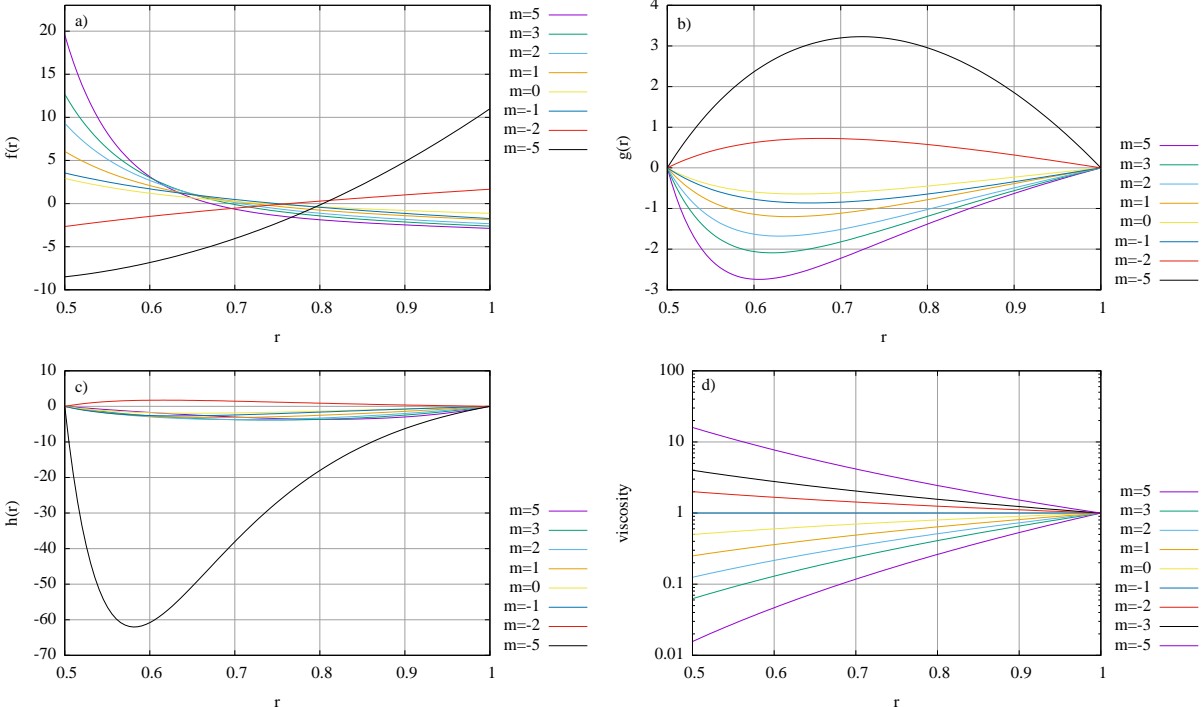

**Figure 1.** Functions $f(r)$ (a), $g(r)$ (b), $h(r)$ (c) and viscosity profile $\mu(r)$ (d) as a function of $r \in [0.5 : 1]$ for various values of $m$.

## 5.2 ELEFANT

ELEFANT borrows largely from the FANTOM code (Thieulot, 2011), but it also brings a number of critical improvements compared to its predecessor, such as spherical shell geometry and the use of a preconditioned conjugate gradient scheme for

both inner and outer iterations (Braess, 2007; Elman, 1996). It is a Finite Element code, based on $Q_1 P_0$ elements which is developed and maintained by the author. It has succesfully been benchmarked against a wide range of analytical problems and also against other codes (including ASPECT) in the case of visco-plastic thermal convection (Tosi et al., 2015).

## 5.3 Setup

In what follows, I set the inner and outer radii to $R_1 = 0.5$ and $R_2 = 1$ respectively. The functions $f(r)$, $g(r)$, $h(r)$ and the

viscosity function are shown in Fig. (1).

  The ASPECT computational grid consists of 12 blocks making up a hollow sphere (Zhong et al., 2000), each block being subdivided in $(2^n)^3$ elements, where $n = 2, 3, ....$ Since elements vary in size in the radial direction, I have chosen to report the average element size in convergence plots, and it is computed as follows:

$$< h >= \left( \frac{V}{N_{el}} \right)^{1/3} \tag{97}$$



where $V$ is the volume of the domain and $N_{el}$ is the total number of elements. The ELEFANT mesh is also based on the same

12 blocks, but each block can be subdivided into $nel^3$ elements where $nel$ is a positive integer wich is not bound to be a power

of 2.

    The analytical velocity solution is prescribed on both internal ($r = R_1$) and external ($r = R_2$) boundaries. Given these

boundary conditions, the pressure is determined up to a constant. Both codes allow for a surface normalisation (the average

25   pressure along the surface has a prescribed value - in this case zero), and a volume normalisation (the average pressure over the

whole volume has a prescribed zero value). Since I have shown here above that both are identically nul for the pressure field,

the choice of pressure normalisation does not matter. Also, the boundary conditions preclude the presence of a pure rotational

mode of numerical origin (Zhong et al., 2008).

    Both codes were run for values of $m = -1$ (constant viscosity) and $m = 3$ (viscosity varies by a factor 16 from the inside to

30   the outside). The density, velocity and pressure fields for both cases are shown in Fig. (2)

    Figure 3 shows the relative root mean square velocity error as a function of the (average) element size for both codes and both

$m$ values. The error is found to quadratically decrease with resolution for both codes. Likewise, the $L_2$-norm of the velocity

error is found to decrease with resolution, linearly for $Q_1P_0$ elements and quadratically for $Q_2Q_1$, as shown in Figs. (4).

Looking at the pressure error convergence, it is found to decrease quadratically with the resolution for both types of elements,

as shown in Fig. (5).

    ELEFANT routinely outputs all three average quantities $<u>$, $<v>$ and $<w>$. All three values were found to be zero

5   within machine precision (oscillating around $10^{-15}$).

## 6   Conclusions

I have derived in this paper a family of analytical solutions to incompressible Stokes flow in a spherical shell under a few

assumptions, such as tangential velocity on the boundaries and a radial viscous profile. The velocity, pressure, density and

viscous fields which satisfy the flow equations at every point in space are then used to benchmark two multi-purpose geo-

10   dynamics codes. $L_2$-norms of the velocity and pressure errors were reported and shown to decrease when the resolution is

increased. Furthermore, various analytical expressions for flow averages were derived and it was shown that the computed

solutions converged to these expected values.

    A number of previous studies (Popov et al., 2014) use an exponential viscosity of the form $\mu(r) = \mu_0 \exp(\boldsymbol{\alpha} \cdot \boldsymbol{r})$ where

$\boldsymbol{\alpha}$ is a parameter controlling the amplitude of the viscosity variations in the system. This approach has been tried during the

preparation of the manuscript, but Eq. (35) then becomes

$$r^2 f'' + (2 + mr)rf' - (2 + mr)f = 0 \tag{98}$$

Although this equation can be solved, the form of the solution $f(r)$ involves the exponential integral function $Ei(r) = -\int_{-r}^{\infty} e^{-t}/t \, dt$ which a) would render the derivations of $g(r)$ and all subsequent quantities very cumbersome, b) would make

5   the solution only semi-analytical. This approach was then abandoned.




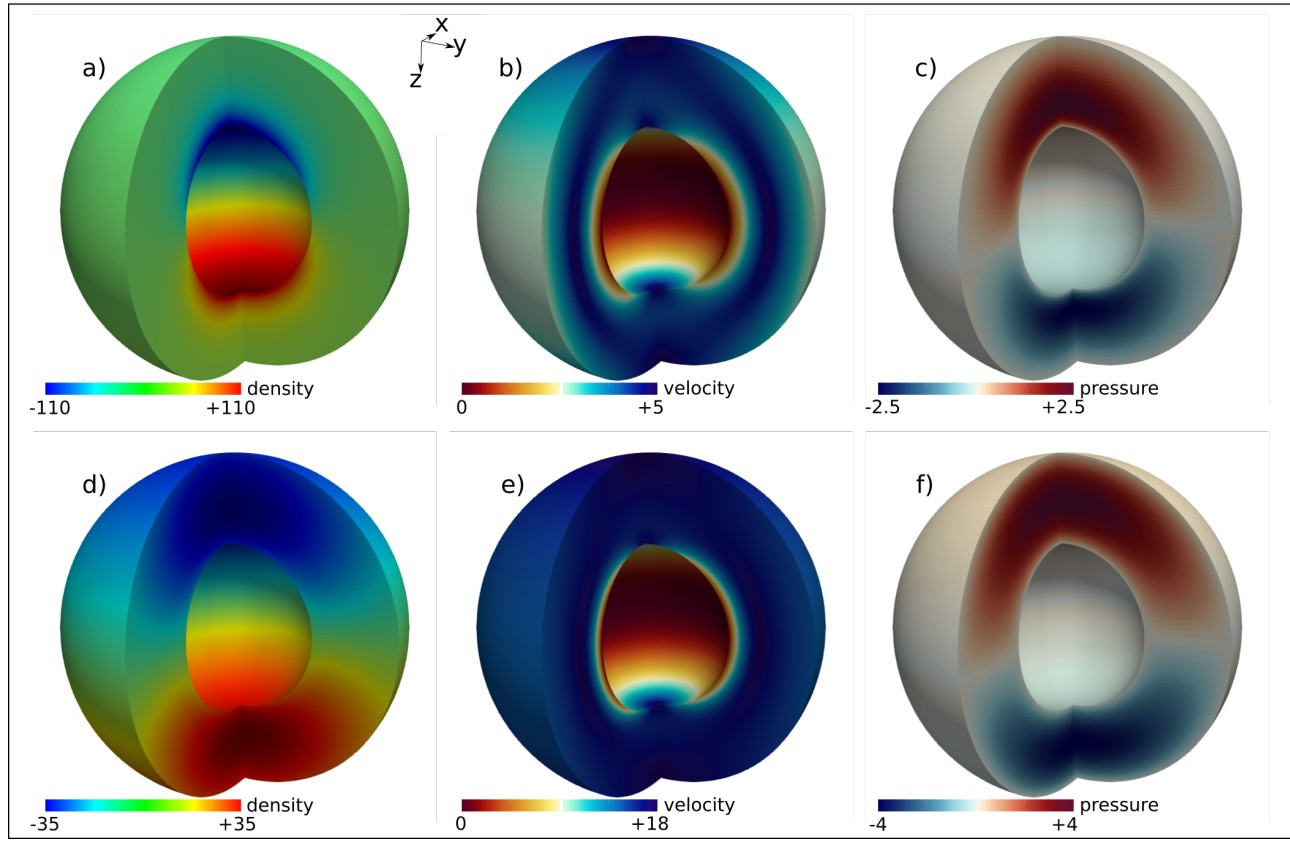

**Figure 2.** a,b,c) Analytical solution for $m = -1$ (constant viscosity); d,e,f) solution for $m = 3$

Looking at the pressure equation, or rather at Fig. (1c), we see that the pressure is zero for $r = R_1$ and $r = R_2$. One simple interpretation is that the pressure in this work should be interpreted as an over pressure with regards to a background lithostatic pressure. Likewise, the (complex) density profiles have to be interpreted as density variations with regards to a background density profile corresponding to the above mentioned lithostatic pressure.

*Author contributions.* All analytical derivations, equations, implementation and numerical simulations have been carried out by the author.

*Competing interests.* The author declares that he has no conflict of interest.





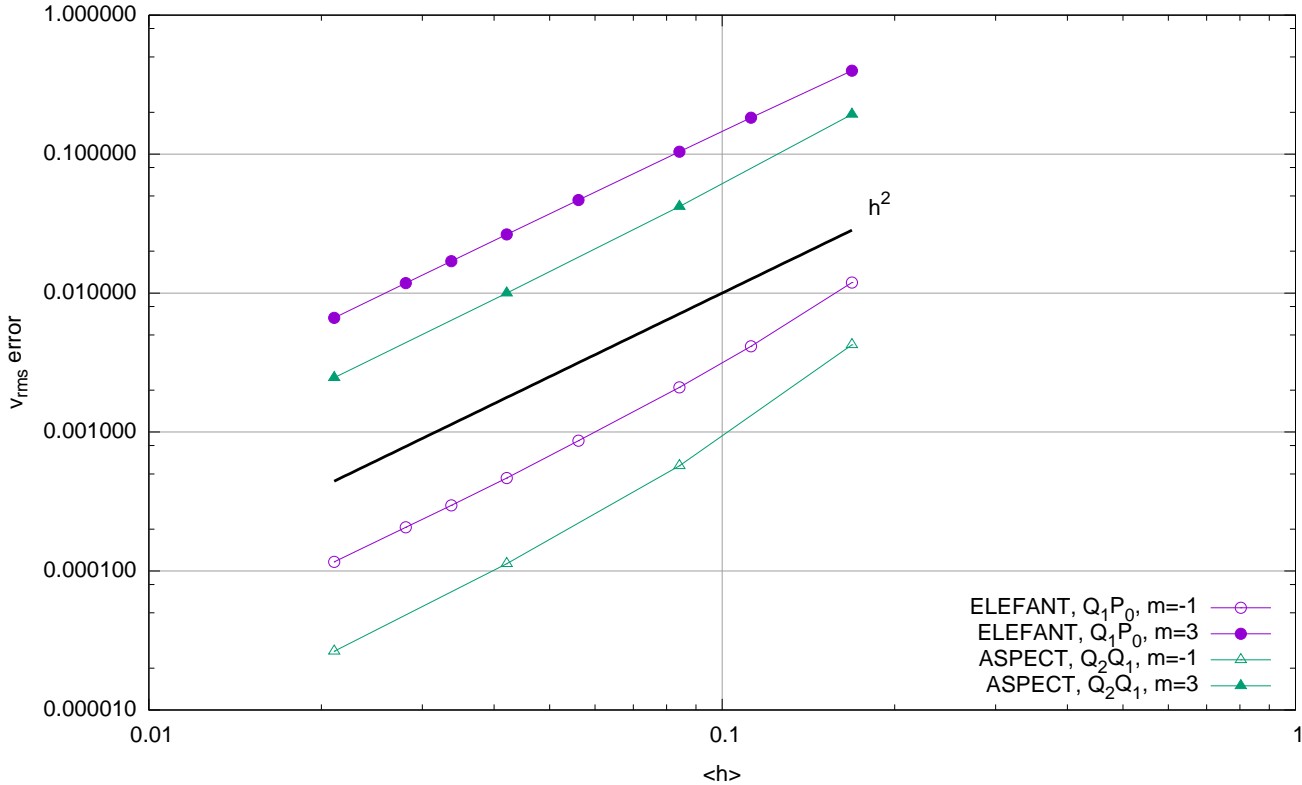

**Figure 3.** Root mean square velocity as a function of the average element size for both codes.

*Acknowledgements.* The author acknowledges stimulating discussions with Prof. G. Puckett in the early stages of this work, and technical
help from A. Glerum and M. Fraters with all things ASPECT. Early reviews by Matt Weller and additional HPC runs by Harsha Lokavarapu
5  are gratefully acknowledged.




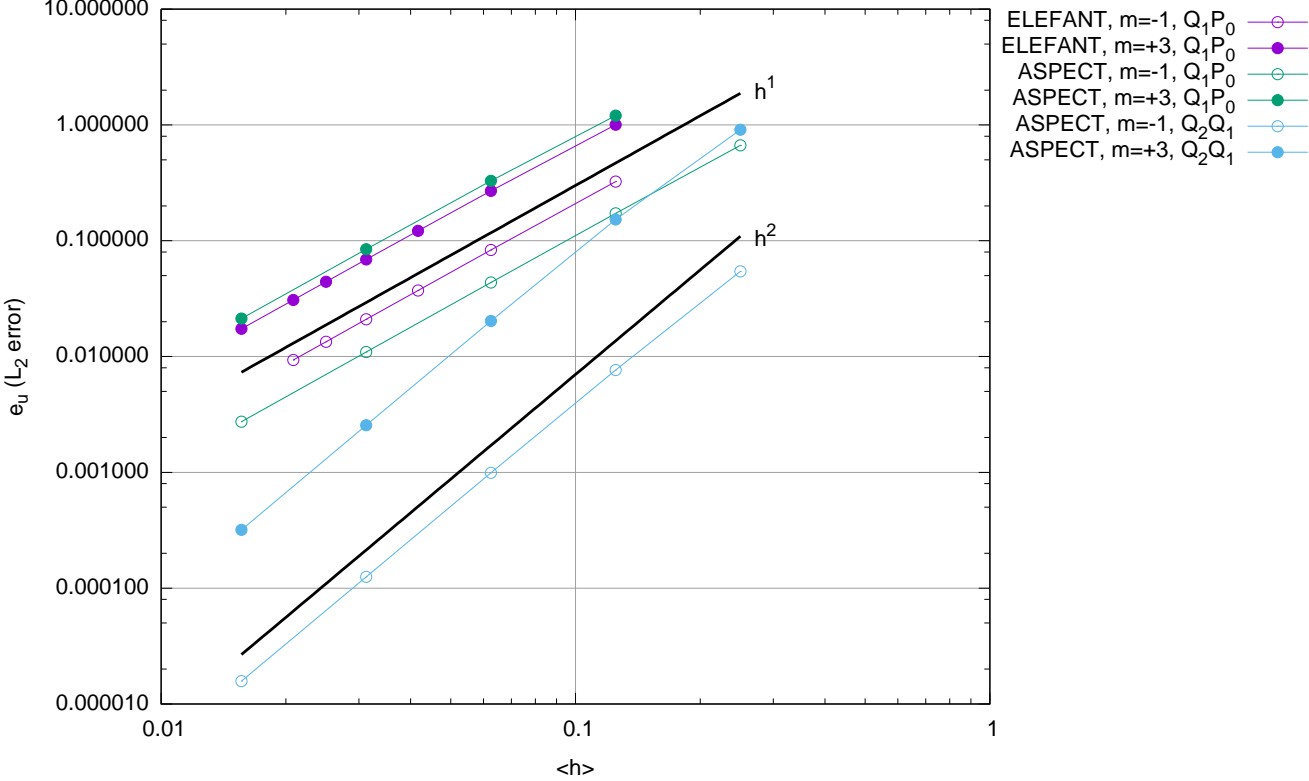

**Figure 4.** Velocity $L_2$-norm error vs average resolution for ELEFANT and ASPECT.

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



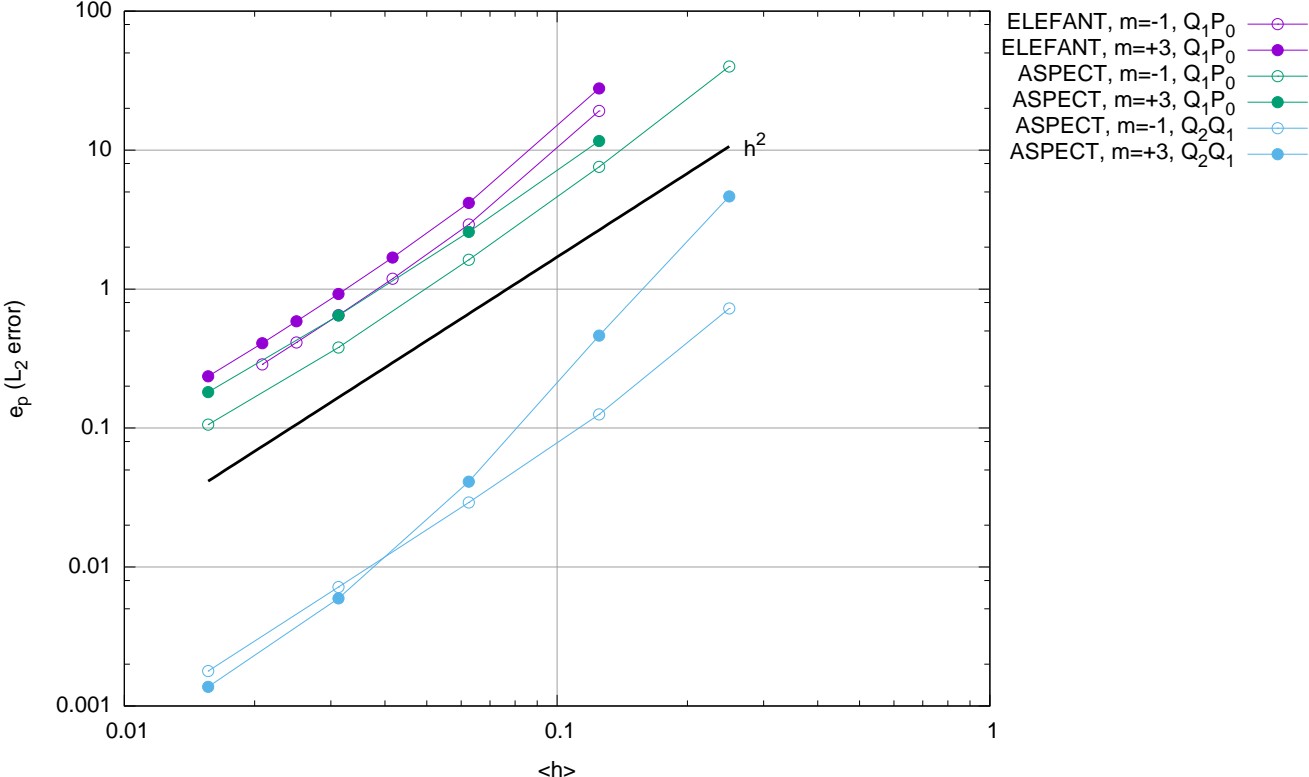

**Figure 5.** Pressure $L_2$-norm error vs average resolution for ELEFANT and ASPECT.

Bull, A., Domeier, M., and Torsvik, T.: The effect of plate motion history on the longevity of deep mantle heterogeneities, Earth Planet. Sci. Lett., 401, 172–182, 2014.

Burstedde, C., Stadler, G., Alisic, L., Wilcox, L., Tan, E., Gurnis, M., and Ghattas, O.: Large-scale adaptive mantle convection simulation, GJI, 192, 889–906, 2013.

5   Busse, F.: Patterns of convection in spherical shells, J. Fluid Mech., 72, 67–85, 1975.

Busse, F. and Riahi, N.: Patterns of convection in spherical shells. Part 2, J. Fluid Mech., 123, 283–301, 1982.

Choblet, G., Čadek, O., Couturier, F., and Dumoulin, C.: OEDIPUS: a new tool to study the dynamics of planetary interiors, Geophy. J. Int., 170, 9–30, 2007.

Dannberg, J. and Heister, T.: Compressible magma/mantle dynamics: 3-D, adaptive simulations in ASPECT, Geophy. J. Int., 207, 1343–1366,
10   2016.

Davies, D., Davies, J., Bollada, P., Hassan, O., Morgan, K., and Nithiarasu, P.: A hierarchical mesh refinement technique for global 3-D spherical mantle convection modelling, Geosci. Model Dev., 6, 1095–1107, 2013.

Elman, H.: Multigrid and Krylov subspace methods for the discrete Stokes equations, Int. J. Num. Meth. Fluids, 22, 755–770, 1996.

Hager, B. and O'Connell, R.: A simple global model of plate dynamics and mantle convection, J. Geophys. Res., 86, 4843–4867, 1981.





Heister, T., Dannberg, J., Gassmøller, R., and Bangerth, W.: High Accuracy Mantle Convection Simulation through Modern Numerical Methods. II: Realistic Models and Problems, 2017.

Kronbichler, M., Heister, T., and Bangerth, W.: High accuracy mantle convection simulation through modern numerical methods , Geophy. J. Int., 191, 12–29, 2012.

Lavecchia, A., Thieulot, C., Beekman, F., Cloetingh, S., and Clark, S.: Lithosphere erosion and continental breakup: Interaction of extension, plume upwelling and melting, Earth Planet. Sci. Lett., 467, 89–98, 2017.

Popov, I., Lobanov, I., Popov, S., Popov, A., and Gerya, T.: Practical analytical solutions for benchmarking of 2D and 3D geodynamic Stokes problems with variable viscosity, Solid Earth, 5, 461–476, 2014.

Richards, M. and Hager, B.: Geoid anomalies in a dynamic Earth, J. Geophys. Res., 89, 5987–6002, 1984.

Stemmer, K., Harder, H., and Hansen, U.: A new method to simulate convection with strongly temperature- and pressure-dependent viscosity in a spherical shell: Applications to the Earth's mantle, Phys. Earth. Planet. Inter., 157, 223–249, 2006.

Tackley, P.: Modelling compressible mantle convection with large viscosity contrasts in a three-dimensional spherical shell using the yin-yang grid, Phys. Earth. Planet. Inter., 171, 7–18, 2008.

Tackley, P.: Dynamics and evolution of the deep mantle resulting from thermal, chemical, phase and melting effects, Earth-Science Reviews,

110, 1–25, 2012.

Thieulot, C.: FANTOM: two- and three-dimensional numerical modelling of creeping flows for the solution of geological problems, Phys. Earth. Planet. Inter., 188, 47–68, 2011.

Thieulot, C.: ELEFANT: a user-friendly multipurpose geodynamics code, Solid Earth Discussions, 6, 1949–2096, 2014.

Tosi, N. and Martinec, Z.: Semi-analytical solution for viscous Stokes flow in two eccentrically nested spheres, Geophy. J. Int., 170, 1015–

20  1030, 2007.

Tosi, N., Stein, C., Noack, L., Huettig, C., Maierova, P., Samuel, H., Davies, D., Wilson, C., Kramer, S., Thieulot, C., Glerum, A., Fraters, M., Spakman, W., Rozel, A., and Tackley, P.: A community benchmark for viscoplastic thermal convection in a 2-D square box, Geochem. Geophys. Geosyst., 16, 10.1002/2015GC005 807., 2015.

van Heck, H., Davies, J., Elliott, T., and Porcelli, D.: Global-scale modelling of melting and isotopic evolution of Earth's mantle: melting

modules for TERRA, Geosci. Model Dev., 9, 1399–1411, 2016.

van Hinsbergen, D., Steinberger, B., Doubrovine, P., and Gassmöller, R.: Acceleration and deceleration of India!Asia convergence since the Cretaceous: Roles of mantle plumes and continental collision, J. Geophys. Res., 116, 2011.

Zhong, S., Zuber, M., Moresi, L., and Gurnis, M.: The role of temperature-dependent viscosity and surface plates in spherical shell models of mantle convection, J. Geophys. Res., 105, 11,063–11,082, 2000.

Zhong, S., McNamara, A., Tan, E., Moresi, L., and Gurnis, M.: A benchmark study on mantle convection in a 3-D spherical shell using CITCOMS, Geochem. Geophys. Geosyst., 9, 2008.