# Peer review of "Analytical solution for viscous incompressible Stokes flow in a spherical shell"

_Solid Earth, 2017_

## Referee Comment (RC1) · Anonymous Referee #1 · 19 Aug 2017

Analytical solutions like those presented in this paper are always useful. I have mostly some minor comments. However, I think that the author must clarify about a number of statements in the paper. By the way, the line numbering in the manuscript does not help for reviewing.

1) Page 2. Referencing convection papers such as van Heck et al. (2016), Dannberg and Heister (2016), and Tackley (2012) for seismic and other structures in the mantle seems a bit odd. Some observational papers are more suitable.

2) Page 2, "... since the analytical solution is not known". This statement is questionable in my view. Spherical harmonic solutions are analytical solutions, and they are not that different from the solutions presented here. They are actually more general, as I would point out later.

[Figure]

3) Page 2, Analytical solutions based on spherical harmonic basis functions have been used extensively in convection benchmark papers like in Zhong et al. [2000; 2008]. However, the author's discussion seems to suggest that they did not.

4) Page 2, "the existing "pure" analytical solutions do not satisfy the condition v*n=0 on the inner and outer boundaries". There are at least two problems with this statement. First, if there was no analytical solution as stated earlier, then how can you talk about "existing" analytical solutions? Second, what do you mean about "pure" analytical solutions? The analytical solutions in Zhong et al [2000; 2008] were all for v*n=0 on the surface and CMB.

Therefore, author's statements as pointed out in comments 2, 3 and 4 are either wrong or inaccurate, and they must be clarified.

5) On equation 17 and the solutions presented here. It seems to me from equation 17 the author only found a special type of analytical solutions that is spherical harmonic degree-1 solution (by assuming solutions as in equation 17. This can be seen from the figures as well. The spherical harmonic solution method can actually be used to construct other forms of solutions, e.g., degree-2, and it is unclear how the author's method can be used for anything other than degree-1 solution. The author should acknowledge this.

---

## Referee Comment (RC2) · Anonymous Referee #2 · 30 Aug 2017

In this manuscript the author presents new analytical solutions for Stokes flow in a spherical shell, which could be used for benchmarking mantle convection codes. This is useful and as his derivation seems correct. I think it can be published with some minor changes to the introductory discussion.

The author writes "Also, the semi-analytical solutions present a major drawback: the solution is given as a function of spherical harmonic expansions which are based on infinite sums and which can be cumbersome to manipulate and/or implement ". This is missing the point somewhat. A great thing about using spherical harmonics is that for azimuthally-constant viscosity the Stokes equation can be solved independently for each spherical harmonic, which means if the driving density field consists of a single spherical harmonic then the flow solution can also be expressed using a single spher-

ical harmonic. No infinite expansion necessary. In fact, this is effectively what the author is doing in this manuscript – his driving density field and flow solution both consist azimuthally of spherical harmonic degree 1 and order 0. Presumably, the solutions the author has found are part of a family of solutions that could also be derived for other spherical harmonics, so if the author wanted to he could generalise this. Another example is that Zhong 2008 presented "analytical" flow solutions for single spherical harmonics (using the propagator matrix method instead of a mathematical equation).

Stylistically, the author has a habit of writing in single-sentence paragraphs, which is normally not considered to be good writing style. In particular, most paragraphs in the Introduction contain only a single sentence. I suggest he groups sentences together to make longer paragraphs. Additionally, it is best to avoid colloquial expressions such as "seen the light of day" in a formal scientific manuscript.

The references the author cites in the first two paragraphs are very recent examples that back up these points; it might be better to cite earlier ones.

The list of "last 30 years" 3D spherical convection codes in the 3rd paragraph is not complete, so he should either insert "e.g." to indicate that these are examples rather than an exhaustive list, or try to make it complete. Missing codes that I can think of right now include the yin-yang grid ones of Kameyama and of Yoshida, and any code that uses a spectral method (codes of Glatzmaier, Zhang+Christensen, Machetel, Monnereau)

One analytic solution the author should mention is Zhong (1996) – although this is not in spherical geometry. It would also be better to mention the Popov et al paper somewhere in the introduction.

Zhong, S. (1996). Analytic solutions for Stokes' flow with lateral variations in viscosity. Geophysical Journal International, 124(1), 18-28.

---

## Author Comment (AC1) · 19 Sep 2017

I wish to thank the reviewers for their positive reactions to the manuscript and to their constructive comments. I list hereafter their original comments (which were rather similar in nature) and my answers to these comments.

Reply to Ref. #1 —————-

1) Page 2. Referencing convection papers such as van Heck et al. (2016), Dannberg and Heister (2016), and Tackley (2012) for seismic and other structures in the mantle seems a bit odd. Some observational papers are more suitable.

-> I have added observational papers and rewrote the first sentence.

[Figure]

2) "... since the analytical solution is not known". This statement is questionable in my view. Spherical harmonic solutions are analytical solutions, and they are not that different from the solutions presented here. They are actually more general, as I would point out later.

-> The sentence was indeed unclear. I have clarified this in the text.

3) Page 2, Analytical solutions based on spherical harmonic basis functions have been used extensively in convection benchmark papers like in Zhong et al. [2000; 2008]. However, the author's discussion seems to suggest that they did not.

-> I have added these references to the list of papers showcasing benchmarks.

4) Page 2, "the existing "pure" analytical solutions do not satisfy the condition v*n=0 on the inner and outer boundaries". There are at least two problems with this statement. First, if there was no analytical solution as stated earlier, then how can you talk about "existing" analytical solutions? Second, what do you mean about "pure" analytical solutions? The analytical solutions in Zhong et al [2000; 2008] were all for v*n=0 on the surface and CMB. Therefore, author's statements as pointed out in comments 2, 3 and 4 are either wrong or inaccurate, and they must be clarified.

-> "pure" is indeed a vague term and I have removed it. I have rewritten this paragraph to clarify my previous statements.

5) On equation 17 and the solutions presented here. It seems to me from equation 17 the author only found a special type of analytical solutions that is spherical harmonic degree-1 solution (by assuming solutions as in equation 17. This can be seen from the figures as well. The spherical harmonic solution method can actually be used to construct other forms of solutions, e.g., degree-2, and it is unclear how the author's method can be used for anything other than degree-1 solution. The author should acknowledge this.

-> I have added a comment at the end of the conclusion section which addresses this

point.

Reply to Ref. #2 ——————-

The author writes "Also, the semi-analytical solutions present a major drawback: the solution is given as a function of spherical harmonic expansions which are based on infinite sums and which can be cumbersome to manipulate and/or implement ". This is missing the point somewhat. A great thing about using spherical harmonics is that for azimuthally-constant viscosity the Stokes equation can be solved independently for each spherical harmonic, which means if the driving density field consists of a single spherical harmonic then the flow solution can also be expressed using a single spherical harmonic. No infinite expansion necessary. In fact, this is effectively what the author is doing in this manuscript – his driving density field and flow solution both consist azimuthally of spherical harmonic degree 1 and order 0. Presumably, the solutions the author has found are part of a family of solutions that could also be derived for other spherical harmonics, so if the author wanted to he could generalise this. Another example is that Zhong 2008 presented "analytical" flow solutions for single spherical harmonics (using the propagator matrix method instead of a mathematical equation).

-> I have added a short comment at the end of the conclusion section which highlights this point.

Stylistically, the author has a habit of writing in single-sentence paragraphs, which is normally not considered to be good writing style. In particular, most paragraphs in the Introduction contain only a single sentence. I suggest he groups sentences together to make longer paragraphs.

-> I have addressed this issue by grouping sentences when applicable.

Additionally, it is best to avoid colloquial expressions such as "seen the light of day" in a formal scientific manuscript.

-> I have removed this expression.

The references the author cites in the first two paragraphs are very recent examples that back up these points; it might be better to cite earlier ones. The list of "last 30 years" 3D spherical convection codes in the 3rd paragraph is not complete, so he should either insert "e.g." to indicate that these are examples rather than an exhaustive list, or try to make it complete. Missing codes that I can think of right now include the yin-yang grid ones of Kameyama and of Yoshida, and any code that uses a spectral method (codes of Glatzmaier, Zhang+Christensen, Machetel, Monnereau)

-> I have added references spanning the last thirty years.

One analytic solution the author should mention is Zhong (1996) – although this is not in spherical geometry. It would also be better to mention the Popov et al paper somewhere in the introduction.

-> Popov et al (2014) and Zhong (1996) are now mentioned in the introduction